# Size–Number and Shape Distribution of Buried Seeds in Soil in a Field Not Cultivated for More Than 10 Years

Luís Silva Dias [†]

Independent Researcher, 7000-713 Evora, Portugal; diasls.pt@gmail.com
† Formerly Department of Biology, Universidade de Évora, 7002-554 Evora, Portugal.

**Abstract:** Seeds act as reserves for plant dispersion in time and their burial in soil plays an essential role in preventing or reducing losses. Two hypotheses regarding the depth distribution of seeds in soil were investigated. One states that the burial of small seeds is restricted to shallower depths than large seeds. The other states that seed shape is important to burial. The fraction of seeds located at depths allowing viable non-photosynthetic growth was also investigated in relation to size and shape. Cores of 20 cm depth were taken from soil with an auger, divided in eight fractions of equal length and sorted through a series of ten sieves, and viable spherical and non-spherical seeds were separately counted. Burial was evaluated by the symmetry of depth–number distributions fitted by Weibull equations. The maximum depth for successful germination and emergence was calculated by combining Weibull equations with published or original material on the relationship between the volume and mass of seeds, and the maximum elongation of hypocotyls in soil. The burial of smaller seeds was found to be restricted to shallower depths, but in larger seeds, size itself appeared to be an unsurmountable barrier to burial. Smaller spherical seeds buried at shallower depths than smaller non-spherical seeds, their number decreasing monotonically with depth, while the number of non-spherical seeds increased from the surface to 10.0–12.5 cm, decreasing thereafter. Larger seeds, spherical or non-spherical, had essentially the same depth–number distribution. In very small seeds ($\leq 0.014$ mm$^3$; approximately 75% of the 29,740 total seeds), almost all spherical and non-spherical seeds were at depths at which non-photosynthetic viable growth would be unsustainable. This fraction reduced as the size of seeds increased, but it never fell below 50% and was only rarely less than 80%. The implications of these high values for aboveground recruitment are discussed in terms of the density of seeds.

**Keywords:** depth–number distribution; non-spherical seeds; precision; seed volume; soil; soil seed banks; spherical seeds; successful emergence

## 1. Introduction

In more than one sense, soils are highly complex systems [1] made up of an intricate web of physical, chemical and biological elements, as well as of processes and organisms that are at the core of agroecosystems and are the support of virtually all life in land ecosystems [2]. The role and importance of plants as primary producers is widely recognized, and their presence aboveground is conspicuous and, therefore, almost universally acknowledged. On the contrary, seeds (in a broad sense, encompassing fruits such as achenes, caryopses and others) from which most plants originate, are out of sight almost everywhere, lying at the soil surface or buried at various depths.

Seeds are viewed as the major reason for the dominance of seed plants [3], which are widely recognized as the most successful plants across the widest range of environments [4]. It is also recognized that, apart from some exceptions, the functioning of plants and plant communities can only be understood if the importance of seeds in soil is acknowledged, because at any given moment, soil seed banks represent the potential population of plants throughout time [5].

Understanding the non-anthropogenic movements and destiny of seeds in soil is an effective way to fulfill this goal. After reaching the soil surface, seeds may germinate, or be eaten, removed or die. The survivors make up the seed bank and can remain at the soil surface or be buried. It has been claimed—frequently without hard evidence being provided—that small and roughly spherical seeds are more easily buried than large seeds, reaching deeper depths, for example, through cracks and crevices [6–8]. The burial of seeds is essentially driven by gravity, directly or indirectly, or by animals burying them, such as earthworms, provided that they feed near the surface and cast deeper in the soil [9–13].

Burial may offer a number of advantages to seeds, increasing persistence and dispersion through time by reducing predation and desiccation, while increasing their chance of finding safe sites with appropriate conditions for germination and survival, including anchorage during germination and the early growth of seedlings [14–16].

Smaller seeds appear to be more or less equally trapped by small and large soil particle sizes across soil surfaces, while larger seeds move unhindered and are only trapped by large soil particles [17]. Smaller seeds will also face higher heterogeneity of soil particles than larger ones [6], implying that they might find more suitable microsites to be retained than larger seeds. In all likelihood, when or where cracks and crevices are not present in soil, the same differential retention of seeds occurs vertically in the soil profile, whereas burial through cracks and crevices might depend upon a variety of processes. These include them being filled with water, in which case surface tension and seed density might play a major role in upward or downward movements, density being a trait that is essentially size-independent, as evidenced in Supplementary Material (Table S1). Therefore, we offer the hypothesis that the non-anthropogenic burial of smaller seeds in soil seed banks is mostly restricted to shallower depths in comparison to the burial of larger seeds. To investigate this and other related or derived hypotheses, soil that was undisturbed for more than ten years was selected and sampled so that the confounding short- and long-term effects of agricultural or pastoral factors were not present.

Investigating size–number distributions in soil seed banks requires determining the size of seeds. This rapidly becomes unfeasible because of the hundreds or thousands of seeds that are easily found when sampling soil, which renders it almost impossible to measure or weigh them all. In practice, passing soil and associated seeds through a series of sieves is probably the only feasible way to derive the size–number distributions of seeds in soil banks, which is necessarily based upon size intervals [18].

Seed shapes represent various trade-offs among forms, such that the efficiency of seed packing, dispersal, landing and seedling establishment is maximized [6]. Nevertheless, the ecological, functional and evolutionary significance of the shapes of seeds has been much less investigated than size. The difficulty of defining and quantifying three-dimensional shape is probably the major reason for the predominance of research on size over shape [19]. Putting aside strictly qualitative approaches, the quantitative descriptions of seed shape usually rely on the population variance of the three orthogonal dimensions of seeds, using the variance as a measure of how much the seed departs from a perfect sphere [7,20]. Again, as mentioned above for size, this would rapidly become unfeasible because of the number of seeds involved, the only practical option being classifying seeds as spherical and non-spherical.

In spite of this, shape, similar to size, may also play a role in seed burial, and spherical seeds might respond to microsite heterogeneity in a way that is similar to smaller seeds, while non-spherical seeds might respond similarly to larger seeds. Despite some conflicting evidence, the persistence and transiency of seeds in the soil seed bank, which might relate to burial, seem to support this reasoning [20,21]. Therefore, we posed the hypothesis that differences in shape distributions in soil seed banks are important for seed burial in the whole range of seed size, with the non-anthropogenic burial of spherical seeds mostly restricted to shallower depths in comparison to the burial of non-spherical seeds.

However, burial may be a two-edged sword if seeds reach such depths that their reserves are insufficient to sustain non-photosynthetic upward growth through soil. Clearly,

the resulting loss of parental investment in seed production would be minimized if burial is restricted in some way, especially for smaller seeds because of the smaller reserves they usually possess. Therefore, we set out to investigate the proportion of spherical and non-spherical seeds located at depths that are capable of allowing seed reserves to sustain non-photosynthetic growth.

## 2. Materials and Methods

### 2.1. Location, Soil Seed Bank Sampling and Sample Processing

A detailed account of the methods followed in this study can be found elsewhere [22] and is summarized in this section. The systematic sampling of soil was performed in mid-winter approximately in the center of a broadly rectangular area with around 27 ha (600 × 450 m) in Mitra Experimental Farm near Évora, Southern Portugal (38°52′93″ N, 8°1′15″ W) in an open *montado* of holm oak (*Quercus ilex* L.) with natural pasture and a gentle and constant slope. At the time of sampling, cultivation or cropping had not been performed on the site for more than 10 years, and sheep grazing had occurred very rarely and with low intensity. The soil was sandy-loam with 50% coarse sand, 24% fine sand, 12% silt and 14% clay.

Three plots of 6 × 2 m$^2$ each and 3 m apart were defined, and three soil cores of 5 cm Ø and 20 cm in depth were taken in each plot at 2 m intervals with an auger that was 30 cm long. Each soil core was divided into eight 2.5 cm fractions starting from the soil surface and sequentially sieved with hand disaggregation under a gentle stream of hot water through a series of ten sieves with a decreasing mesh size: 2.38, 0.85, 0.71, 0.56, 0.425, 0.355, 0.297, 0.25, 0.212 and 0.149 mm.

The fractions retained by the sieves of a 0.85 mm or lower mesh size were examined under a stereomicroscope, while those retained by the 2.380 mm mesh size were examined by the naked eye. Due to the very high number of seeds, seed shape classification as spherical (or roughly spherical) or clearly non-spherical was carried out visually and will hereafter be referred to as spherical and non-spherical seeds, respectively. Seeds were considered viable if they were intact at sight and resisted to a moderate pressure with tweezers [23]. Viable seeds were sorted and counted according to their shape. Raw data are available as Supplementary Material (Table S2) in a non-proprietary comma-separated values (CSV) format file.

No attempt was made to identify the species of each and every seed, but only the identification of species found in the samples. The details of species identification and a complete list of the 20 species found in the seed bank, mostly annuals, can be also found in [22].

### 2.2. Modeling Size–Number Distributions and Data Analyses

The distributions of spherical and non-spherical seeds to depth were fitted by the two-parameter Weibull function [24]. Details on the models investigated to describe the depth–number distributions of seeds and the subsequent selection of the two-parameter Weibull function can be found in Appendix A. The Weibull equations were fitted to all classes of sizes combined and separately to each class of size by least squares nonlinear regression without replication using the Marquardt method [25].

The two-parameter Weibull function can be expressed as:

$$Y = 1 - \exp - [(X/b)^c], \tag{1}$$

where $Y$ is the cumulative abundance of seeds at depth $X$ in proportion to the total number of seeds in all classes of depth; $b$ is a scale parameter estimating the depth at which 63% of seeds are found; $c$ a dimensionless shape parameter estimating the symmetry of the distribution of seeds over soil depth, with $3.25 \leq c \leq 3.61$ showing symmetry and representing a good approximation to the normal distribution, $c < 3.25$ positive asymmetry, $c > 3.61$ negative asymmetry [26,27]. Equations were only accepted after a consistency check of the parameter estimates and the predictions of seeds frequency against the original data.

Casco et al. [18] showed that the mesh size is an appropriate surrogate for the seed size, preferable to mesh bisector, and that the mesh size could be equated to the geometric mean of the length, width and thickness of seeds assumed as ellipsoids revolving around length. Therefore, the volume of seeds (*VOL*) can be deduced from the mesh size (*MS*) as:

$$VOL = \pi\, MS^3/6, \tag{2}$$

and the size of seeds be expressed as interval. For example, seeds retained by the mesh size of 0.149 mm will be denoted as [0.002, 0.005] mm$^3$, 0.002 mm$^3$ and 0.005 mm$^3$ being the volumes of seeds with a geometric mean of a length, width and thickness of 0.149 mm and 0.212 mm, respectively.

Values of seed mass were calculated from the volumes deduced from the mesh size (Equation (2)) using the equations fitted by Clifford [28], Sánchez et al. [29], and from the allometric equation fitted as described in Appendix B:

$$M = 0.521\, VOL^{1.081}, \tag{3}$$

where *M* is seed mass (mg) and *VOL* is seed volume (mm$^3$). The maximum depth for successful germination in soil was estimated from the mean depths derived from the allometric equation of Bond et al. [30] and from the "one site" binding hyperbola of Benvenuti and Mazzoncini [31], using as input the average mass of the estimates provided from the equations published by Clifford [28], Sánchez et al. [29], and from Equation (3).

Combining the maximum depth for the successful germination of seeds in soil with the Weibull equations fitted to depth–frequency data, it was possible to estimate the percentage of seeds for which the size, as a surrogate of the reserves of seeds, could successfully sustain non-photosynthetic growth (*SG*).

The data of Weibull equations including $R^2$ and the derived *SG*-values are available as Supplementary Material (Table S3) in a non-proprietary comma-separated values (CSV) format file.

The relative precision of the means (*D′*) of the number of seeds, of *c* and of *SG*, was determined from the half-length of the $1 - \alpha$ confidence interval of the means as:

$$D' = t\, \mathrm{SE}/\overline{Y}, \tag{4}$$

where *t* is the value of Student's *t* distribution with $n - 1$ degrees of freedom and a type I error probability $\alpha$, here set at 0.05, and SE is the standard error of the mean $\overline{Y}$ [32,33]. The data of *D′* are available as Supplementary Material (Table S4) in a non-proprietary comma-separated values (CSV) format file.

Comparisons between the *c*- and *SG*-values of the spherical and non-spherical seeds were made by two-tailed paired Student's *t* tests with comparison-wise type I error rates of 0.05.

The data are presented as mean $\pm$ SE. Linear and non-linear regressions were performed with Statgraphics 4.2 (STSC, Inc., Rockville, MD, USA). All other statistics were obtained with Excel® 2010 (Microsoft Corporation, Redmond, WA, USA).

## 3. Results

The total number of seeds was 29,740, 47% being spherical (13,930) and 53% non-spherical (15,810). The distribution of the number of seeds arranged by depth (all sizes combined) and by size (all depths combined) are presented in Figure 1.

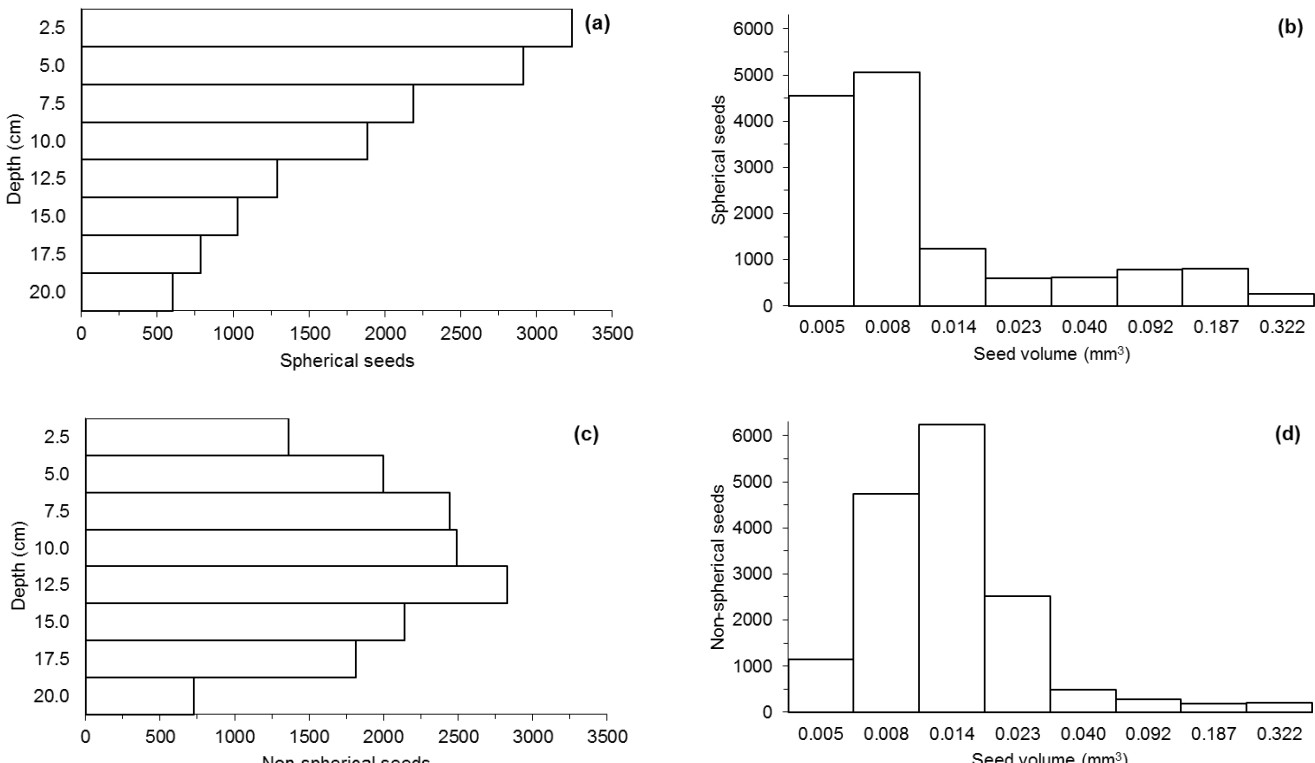

**Figure 1.** Number of spherical (**a,b**) and non-spherical (**c,d**) seeds in relation to soil depth, all sizes combined (**a,c**), and to seed volume, all depths combined (**b,d**). The volume of seeds refers to the highest value in each interval resulting from the sequential sieving.

The relative precision of the means of the number of seeds arranged by shape and size (Table S4) ranged from 0.201 to 2.131 (0.248 to 1.411 in spherical seeds; 0.201 to 2.131 in non-spherical seeds). The mean value of $D'$ was 0.608 ± 0.134 (0.566 ± 0.137 and 0.657 ± 0.252 in spherical and non-spherical seeds, respectively).

Equation (1) could always be fitted to the depth–number distributions of the two shapes combined, and of spherical and non-spherical seeds separately. The adjusted coefficients of determination (Table S3) ranged from 0.973 to ≈1 (0.973 to ≈1 in spherical seeds; 0.981 to ≈1 in non-spherical seeds). The mean value of $R^2_{aj}$ was 0.995 ± 0.0005 (0.994 ± 0.0007 in spherical and 0.995 ± 0.0005 in non-spherical seeds, respectively).

The values of the shape parameter $c$ of Equation (1) (Table S3) ranged from 0.695 to 3.487 (0.695 to 3.487 in spherical seeds; 1.014 to 3.187 in non-spherical seeds). In all samples except one, with spherical seeds with the size [0.014, 0.023) mm$^3$, the value of $c$ was outside the interval that reveals symmetry. The mean value of $c$ was 1.739 ± 0.046 (1.541 ± 0.057 and 1.996 ± 0.059 in spherical and non-spherical seeds, respectively). The mean values of $c$ are presented in Figure 2.

The relative precision of the mean values of $c$ arranged by shape and size (Table S4) ranged from 0.084 to 0.747 (0.084 to 0.747 in spherical seeds; 0.125 to 0.528 in non-spherical seeds). The mean value of $D'$ was 0.260 ± 0.046 (0.261 ± 0.074 and 0.260 ± 0.056 in spherical and non-spherical seeds, respectively).

The differences between the $c$-values of seeds of all sizes combined were investigated by two-tailed paired comparisons, and were found to be significant ($t_8 = 11.921$, $p = 2.255 \times 10^{-6}$), as were the differences between spherical and non-spherical seeds of sizes [0.002, 0.005) mm$^3$ ($t_7 = 5.726$, $p = 0.001$); [0.005, 0.008) mm$^3$ ($t_8 = 7.283$, $p = 8.527 \times 10^{-5}$); and [0.008, 0.014) mm$^3$ ($t_8 = 3.658$, $p = 0.006$). When significant differences were found, the means of $c$ in spherical seeds were always lower than in non-spherical seeds of the same size. Conversely, for all seed sizes larger than 0.014 mm$^3$, no significant differences were found ($p \geq 0.111$).

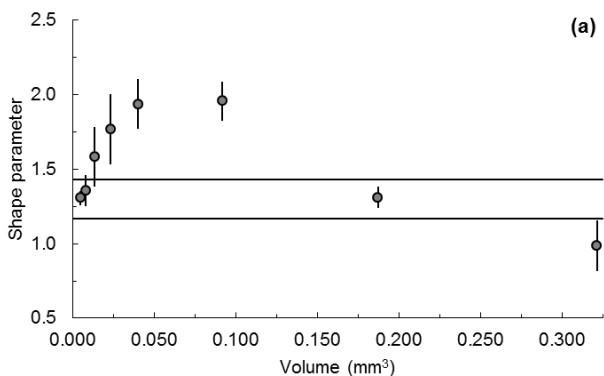
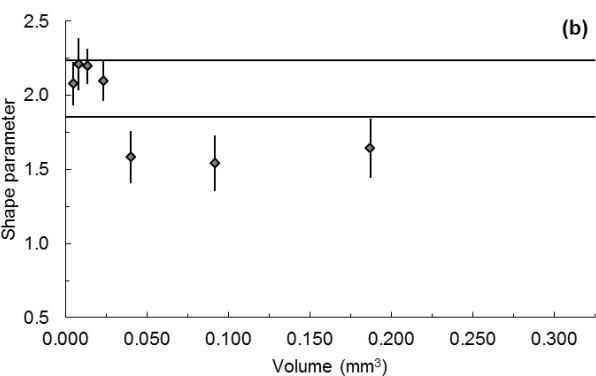

**Figure 2.** Mean ± SE of shape parameter c in relation to seeds size. Horizontal lines show the 95% confidence intervals of the means of all sizes combined: (**a**) spherical seeds; (**b**) non-spherical seeds. Volume of seeds refers to the highest value in each interval resulting from the sequential sieving.

The maximum depth estimated for the successful germination of seeds in soil ranged from 2.6 mm for the lowest class of seed size [0.002, 0.005) mm$^3$ to 26.6 mm for the largest class of spherical seeds [0.187, 0.322) mm$^3$, and 18.8 mm for the largest class of non-spherical seeds [0.092, 0.187) mm$^3$ (Table S3). Replacing $X$ in Equation (1) by the maximum depths of each range of volumes resulted in $SG$ values estimating the percentage of seeds for which successful germination would be possible.

In spherical seeds, the $SG$ ranged from less than 5% in seeds <0.092 mm$^3$ to a maximum of 50.0% in seeds of [0.187, 0.322) mm$^3$, with a mean value of 4.4 ± 1.0%. In non-spherical seeds, the $SG$ was almost always lower than 10%, and never exceeding 21.0% (Table S3). The mean values of $SG$ are presented in Figure 3.

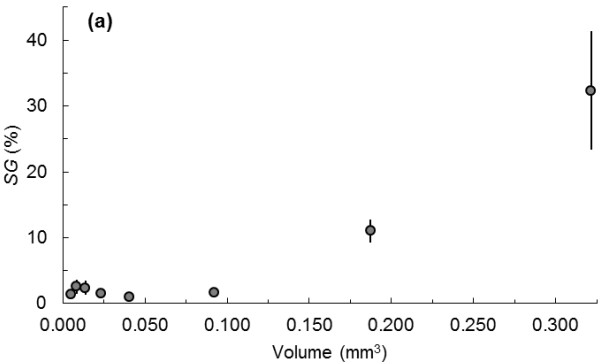
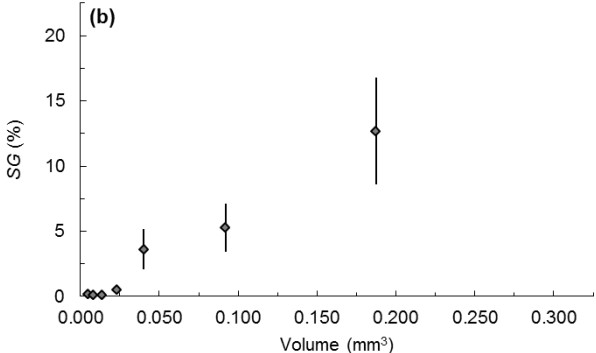

**Figure 3.** Mean ± SE of the percentage of seeds for which successful germination would be possible given their size ($SG$): (**a**) spherical seeds; (**b**) non-spherical seeds. The volume of seeds refers to the highest value in each interval resulting from the sequential sieving.

The relative precision of the means of $SG$ arranged by shape and size (Table S4) ranged from 0.367 to 1.428 (0.367 to 1.200 in spherical seeds; 0.655 to 1.428 in non-spherical seeds). The mean value of the relative precision of $SG$ was 0.939 ± 0.081 (0.830 ± 0.106 and 1.063 ± 0.112 in spherical and non-spherical seeds, respectively).

Differences between $SG$ were investigated by two-tailed paired comparisons and were found to be significant for the seeds of sizes [0.002, 0.005) mm$^3$ ($t_7 = 5.527$, $p = 8.808 \times 10^{-4}$) and marginally for [0.005, 0.008) mm$^3$ ($t_8 = 2.313$, $p = 0.049$), with the means of spherical seeds always higher than those of non-spherical seeds. Conversely, for all seed sizes larger than 0.008 mm$^3$, no significant differences were found ($p \geq 0.061$).

## 4. Discussion

The abundance of seeds in soil banks is notoriously variable even at small distances [34] and the data reported here are no exception. Keeping in mind that the precision of means

varies inversely with the value of $D'$, the precision of the means of the number of seeds was low, with $D'$ expressed in terms of the half-length of the confidence interval reaching as high as 2.131 and never falling below 0.201.

However, a completely different picture emerges in terms of the symmetry $c$ of the depth–number distribution. The precision increases drastically, never exceeding 0.747, and on average the $D'$-values are 2.5 times lower than those of the number of seeds. Symmetry shows evidence of being essentially inelastic and independent of the actual number of seeds, basically depending upon the size and shape of seeds. The consistency of the pattern of $c$ supports the use of this parameter whenever a description of the depth–number distributions of seeds in soil is desired.

Conversely, the percentages of seeds for which successful germination might be viable (*SG*) are even less precise than the number of seeds, and on average their $D'$-values are double those of the number of seeds and fourfold those of symmetry $c$.

### 4.1. Is the Burial of Small Seeds Restricted to Shallower Depths Than that of Large Seeds?

Before initiating the discussion of this hypothesis, a note on the meaning and definition of what is a small or a large seed is required. In fact, there is no such definition, at least of a quantitative kind. In addition, some of the few attempts to numerically separate small from large seeds refer to weight and not volume, setting the boundary between small and large at 0.5 mg [35] to 1.5 mg [36]. An exception, in which small and large seeds are distinguished using only one size measure, places the boundary at 1–2 mm of length [37]. Either way, all seeds recorded in the investigation presented here would be classified as small. Therefore, caution must be exerted, and it must always be kept in mind that small and large are to be viewed in relation to the actual range of volumes in the study, which may be a reflection of the community studied [38]. In this case, it extends by three orders of magnitude, from 0.002 mm$^3$ to 7.059 mm$^3$ in spherical seeds, and very rarely to more than 7.059 mm$^3$ in non-spherical seeds. As a guide in the discussion of the results presented here, the cut-off for very small seeds will be placed at 0.014 mm$^3$, for small seeds at 0.040 mm$^3$ and for intermediate to large seeds at or above 0.092 mm$^3$.

The number of seeds (all sizes combined) was approximately equal in all depth intervals down to 12.5 cm, with 76% of seeds accumulated, while seeds up to 0.008 mm$^3$ (all depths combined) accounted for 77%, suggesting that small seeds were restricted to shallower depths, as predicted by our first hypothesis.

However, differences between spherical and non-spherical seeds are evident. The number of spherical seeds (all sizes combined) reduces monotonically with depth, while the number of non-spherical seeds increases down to 12.5 cm, decreasing thereafter. Similarly, the size–number distribution of spherical seeds is highly asymmetrical to the left with a very pronounced tail to the right, while in non-spherical seeds the same type of asymmetry occurs but is less intense. It is worth noting that despite these differences, size–number distributions were previously found to be essentially non-random and governed by short-term and short-range factors regardless of shape [22].

Considering the above, a more detailed look at the differences in the depth distribution between spherical and non-spherical seeds is required, and for this the shape parameter of the Weibull function is an invaluable tool, even more so because of its high precision.

The shape parameter $c$ of the Weibull function (Equation (1)) measures the symmetry of the depth–number distribution of seeds. With only one exception, its value was always lower than the lowest value that indicates symmetry. For all practical purposes, depth–number distributions were always positively asymmetric.

In positively asymmetrical distributions, for which the log-normal distribution is the best-known representative, the factors that govern distributions are few and act multiplicatively [39]. In this case, it shows that the greatest part of seeds is located at shallower depths, increasingly near the soil surface as the value of $c$ decreases.

Despite the fact that spherical and non-spherical seeds of all sizes combined were located at shallow depths, the former were located at significantly lower depths than the latter.

However, as the size increased the value of *c* increased or remained essentially unchanged, meaning that seeds were mostly located deeper or at the same depth, depending on their shape, until some threshold value of volume was reached, which is larger in spherical seeds, and at which the value of *c* falls abruptly, meaning that the burial of the majority of larger seeds was restricted, and they were located at shallow or very shallow depths.

Regardless of the driving force behind seed burial, the hypothesis that the burial of smaller seeds is restricted to shallower depths in comparison to larger seeds is essentially supported by our data. This is not to say that small seeds cannot reach the same or greater depths in soil than large seeds, and our data show that they can, especially spherical seeds. However, the issue is not so much whether some small seeds can reach the same or greater depths than large ones, which they do, but whether they do so in a proportion that is consistent with the observed small/large seed ratio, or if, on the contrary, despite being able to bury deep, they are located preferably at lower depths. The analysis of the shape parameter *c* substantiates this preferential location at lower depths, and therefore, supports the hypothesis that the burial of non-germinated smaller seeds is restricted to shallower depths.

However, this fact is affected by seed shape, and we must acknowledge that there is a volume of seeds at which the seeds not only stop burying, but on the contrary, size itself is a barrier to burial. This allows the burial of most of the large seeds only at very shallow depths, even shallower than that achieved by small seeds. In all likelihood, this is not because they might be trapped by soil particles [17] or meet lesser heterogeneity of soil particles [6], but because their size, probably aggravated by shape in non-spherical seeds, might be too large to find continuous paths allowing downward movements.

We must also consider the type of soil from which the samples were taken, and how close the sampled soil was to an ideal non-anthropogenically disturbed soil, which not only affects the size–number distribution across the soil profile, as discussed in this section, but also the size–number distribution of the shapes and fractions of seeds located at depths allowing plantlets to reach the soil surface, as discussed next.

Regarding the first issue, it has been reported that the type of soil may differentially affect the burial of small and large seeds, and of spherical and non-spherical seeds [31]. Therefore, the results presented here for a sandy-loam with the characteristics of the sampled soil may not hold fully in other soils, and more research is undoubtedly required so that a more complete picture might emerge. As for the second issue, concerning the closeness of the soil sampled to an ideal non-anthropogenic disturbed soil, the body of research involving rates of soil recovery is relatively scarce [40], a fact that is aggravated by the lack of information on the loads imposed before the fallow, which lasted for at least ten years before seed bank sampling. However, evidence exists [40–42] that ten years without cultivation might be sufficient to result in soil characteristics that are essentially similar to those found in an ideal non-anthropogenically disturbed soil, including those characteristics more likely to impact seed burial.

*4.2. Are Differences in Shape Important for Seed Burial across the Whole Range of Seed Size?*

In the preceding section, the importance of seed shape for seed burial emerged repeatedly, supporting the hypothesis that differences in seed shape, visually classified as spherical and non-spherical, are important for seed burial, but not necessarily across the whole range of seed sizes observed. All the same, a quick review of the evidence is worthwhile.

The most of the spherical seeds were found at or above 7.5 cm, while the most part of the non-spherical seeds were found from 5 cm to 15 cm. In addition, the spherical seed numbers decreased monotonically with depth, while non-spherical seeds numbers increased from the surface down to the interval 10.0–12.5 cm, decreasing thereafter.

All sizes considered, most spherical seeds were buried at shallower depths than most of the non-spherical seeds, as shown by the mean value of the shape parameter *c* with all sizes combined. The same occurred when smaller seeds were considered, an effect that

tended to disappear in larger seeds, wherein spherical and non-spherical seeds presented essentially the same depth–number distributions.

A final remark. The shape of seeds is usually classified in a continuum of deviations from sphericity [7,19–21] and not as the dichotomy spherical/non spherical that had to be followed here. This binarization was a practical and unavoidable consequence of the objectives and design of the study but did not prevent the detection of differences between the shapes defined here as spherical and non-spherical.

### 4.3. What Fraction of Seeds Is Located at Depths Allowing Plantlets to Reach the Soil Surface?

Seed burial presents a number of adaptive advantages—namely, escape from herbivores or surface-foraging predators [43] and the avoidance of wetting-desiccation cycles leading to unsustainable seed germination—which are felt more intensely at the soil surface than belowground [44]. Additionally, buried seeds may face a top-down resistance imposed by soil that facilitates root penetration downward, while the resistance faced by the elongating roots of seeds germinated at the soil surface may prevent them from penetrating, pushing seeds away from the surface, and eventually leading to plant death by desiccation [14]. At the other end, burial can take seeds to such depths that, regardless of whether environmental cues for germination to start are active, their reserves are insufficient to sustain non-photosynthetic growth until the plantlet emerges through the soil surface and starts photosynthesizing.

Before delving further into the discussion of the results, a note of caution is due. Contrary to the symmetry parameter $c$ of the Weibull equations, the precision of the means of the percentages of seeds located at depths allowing successful germination and emergence ($SG$) was low—even lower than the precision of the number of seeds. For reasons presented in Appendix A, samples were sometimes removed from the analysis and sample sizes turned out to be smaller than intended in the sampling design. This was particularly true for the largest seed sizes. However, no relationship between precision and sample size was observed. Naturally, more research on this matter is needed, but the results obtained suggest that the very low precision observed may essentially be due to the very high small-scale spatial variability known to occur in soils.

For small seeds, spherical and non-spherical alike, and for spherical seeds of intermediate size, virtually all were located at depths at which successful germination would not be viable, with significantly smaller fractions of seeds located at favorable depths in very small non-spherical seeds, which were mostly located deeper than spherical ones.

Differences between spherical and non-spherical seeds were again evident. Besides the higher values of $SG$ for small spherical seeds at favorable depths, those percentages remained essentially constant in a very narrow range between 1% and 3% up to sizes < 0.092 mm$^3$, whereas in non-spherical seeds, $SG$ values remained essentially unchanged in a very narrow but lower range between 0.1% and 0.5%, and only up to sizes < 0.023 mm$^3$, thereafter increasing rapidly, largely surpassing the $SG$ values of spherical seeds.

Very small seeds, here defined as those with a size smaller than 0.014 mm$^3$, represent slightly more than 75% of spherical and non-spherical seeds. Most of the former (89% to ≈100%) and virtually all of the latter (99% to ≈100%) were located at depths at which non-photosynthetic growth would be unsustainable. As could be expected, a trend towards a reduction in these percentages is evident as size increases, but even so, in the largest spherical seeds, the percentages still reached 50 to 80%, while in the largest non-spherical seeds they were 79 to 92%.

The sampling design accounted for the practical impossibility of individually determining the size of seeds and required the use of intervals of size. The determination of the proportion of seeds located at favorable depths was based upon the largest possible seed size in each interval. Had it been based upon the smallest, the percentage of seeds at favorable depths would necessarily be lower.

After reaching such unfavorable depths, seeds might decay, be eaten or germinate and the resulting plants die after exhausting their reserves before being able to photosynthesize,

in all cases depleting the seed bank; alternatively, they may be moved upwards to favorable depths by earthworms casting at or near the soil surface [10,45], by flooding [16] or by the removal of top soil by erosion. They may also stay at unfavorable depths, giving rise to a persistent fraction of soil seed banks that results from their location in the soil profile.

Conversely, those that remain at favorable depths constitute an intrinsically persistent fraction of soil seed banks, in the sense that they could successfully germinate in the future without any change of depth. According to the data reported here, such persistence would be more associated with spherical seeds, especially smaller ones, while the lack of persistence would be more characteristic of non-spherical seeds, especially again smaller ones. In general terms, this scheme overlaps very well with the classic scheme set out by Thompson et al. [20] to predict persistence in soil by size and shape in northwest Europe. Surprisingly, the overlap was stronger in fruits than in seeds, although the diaspores of three quarters of the species found in this study were not fruits but seeds.

The percentage of seeds located too deep for their size to allow successful germination implies a loss of parental investment, and unless they move or are moved upwards it is doubtful that the intrinsically persistent fraction of the seed bank could in fact allow the reconstitution of communities in extreme cases of catastrophic events completely eliminating aboveground vegetation.

To try to answer this question, it is necessary to abandon percentages, return to total numbers and estimate for all sizes the number of seeds per unit of area buried at depths allowing viable germination. In this case, it was slightly more than two spherical seeds per cm$^2$, and about 0.3 non-spherical seeds per cm$^2$. Extrapolating to a larger area, we obtain more than 22,000 and more than 3000 seeds per m$^2$, respectively. Even accounting for dormancy and mortality, in all likelihood, these densities would allow the restoration of aboveground vegetation even in the extreme case of its complete elimination before the production of new seeds.

## 5. Conclusions

After reaching the soil surface, viable seeds may germinate, die, be eaten or lost, or join the seed bank and stay at the surface, or be buried in the soil. The burial in the soil of smaller seeds was found to be restricted to shallower depths than the burial of larger seeds, and even though the former can reach the same or deeper depths than the latter, smaller seeds locate preferentially at shallower depths. However, very large seeds, especially non-spherical ones, because of their size and shape, tend to be located at depths that are even shallower than smaller seeds.

Spherical and non-spherical seeds differed in all the parameters examined, differences that reduced in larger seeds.

Regardless of shape, all or almost all small seeds were located at depths at which successful germination and emergence would be impossible, suggesting that, in addition to the recognized physiologically mediated persistence of seeds in soil, a positional persistence should also be considered in ecological reasoning regarding soil seed banks. It is not surprising that the fraction of seeds buried at depths not hindering the recruitment of new plants increased with size. Altogether, and even in the absence of upward movements of seeds in soil, the small fraction of seeds whose size would not prevent the emergence of new plants may be enough to restore aboveground vegetation, even in the case of complete elimination before new seeds are formed.

**Supplementary Materials:** The following supporting information can be downloaded at: https://www.mdpi.com/article/10.3390/ijpb13040039/s1, Table S1: Volume (mm$^3$) and mass (mg) of seeds; Table S2: Number of spherical and non-spherical seeds by mesh size and soil depth; Table S3: Parameters and coefficients of determination adjusted for degrees of freedom of Weibull equations (Equation (1)), maximum depths for successful germination and percentages of seeds for which successful germination would be possible (*SG*) arranged by sample, shape and size; Table S4: Relative precision of the mean (*D*′) of number of seeds (*N*), symmetry of depth–number distribution (*c*), and

percentage of seeds for which successful germination would be possible given their size (*SG*) in spherical and non-spherical seeds sorted by size.

**Funding:** This research was funded by European Community under the project EC AIR-CT-920029.

**Institutional Review Board Statement:** Not applicable.

**Informed Consent Statement:** Not applicable.

**Data Availability Statement:** The data presented in this study are available as Supplementary Materials to this paper.

**Acknowledgments:** The author wishes to thank Maria Gertrudes Grenho (Department of Biology, University of Évora, Portugal) for her help in measuring and weighing seeds.

**Conflicts of Interest:** The author declares no conflict of interest. The funder had no role in the design of the study; in the collection, analyses, or interpretation of data; in the writing of the manuscript, or in the decision to publish the results.

## Appendix A

Considering the ten classes of seed size resulting from the ten sieves used in the sequential sieving of soil cores plus the additional class resulting from combining the seeds retained in all sieves, the three soil cores taken in each of the three plots, and the two shapes, the total number of samples was 198. However, in some samples no seeds were found. For example, spherical seeds were never found in the sieve with a mesh size of 2.38 mm and non-spherical seeds were only found once in that sieve. Similarly, seeds were frequently not found in sieves with a mesh size of 0.85 mm.

Therefore, the spherical and the non-spherical seeds found in the sieves with a mesh size of 0.71 mm or higher (when any) were combined, reducing the number of samples to 162. However, in a number of cases, seeds were also absent in sieves with a mesh size of 0.71 mm, thus, further reducing the number of samples.

In addition, in a number of samples, seeds were also absent in intermediate soil depths. The implication is that fitting equations of cumulative seed proportion to soil depth would necessarily result in non-null estimates of seed numbers when in fact there were none. Such samples were also discarded.

Altogether, the final number of samples to fit depth–number distributions reduced to 133; 75 of spherical seeds and 58 of non-spherical seeds.

A graphical analysis of data suggested that the cumulative distribution of seeds could be satisfactorily described by the three-parameter Weibull function, or by the fourth-degree polynomial with the frequency of seeds logarithmically transformed or left untransformed.

The three-parameter Weibull function [24] can be expressed as:

$$Y = 1 - \exp - \{[(X - X_0)/b]^c\}, \tag{A1}$$

where $Y$, $X$, $b$ and $c$ have the same meaning as in Equation (1), and $X_0$ is the location parameter estimating the depth at which the frequency of seeds is strictly non-null. In this form, the estimate of the depth at which 63% of seeds are located is given by $X_0 + b$. The Weibull equations were fitted as described above in Section 2.2 and, likewise, equations were only accepted after a consistency check of the parameter estimates and predictions of seeds frequency against the original data.

The fourth-degree polynomial can be expressed as:

$$Y = a_0 + a_1 X + a_2 X^2 + a_3 X^3 + a_4 X^4, \tag{A2}$$

where $Y$ and $X$ have the same meaning as in Equations (1) and (A1). In addition, the fourth-degree polynomial was also investigated with the cumulative frequency of the seeds transformed as $\ln Y$. Equation (A2) was fitted separately using $Y$ and $\ln Y$ by forward stepwise least squares linear regression without replication with the comparison-wise error rate for coefficients set at 0.05. Equations were only accepted after a consistency check of

the parameter estimates and seeds frequency predictions, including the requirement that the predicted cumulative frequency of seeds could never decrease with depth.

Equations were fitted to a subset corresponding approximately to 15% of all samples, with samples of the spherical and the non-spherical seeds represented approximately in the same proportion. The subset was randomly selected and comprised 11 samples of spherical seeds and nine of non-spherical seeds.

While fitting the three-parameter Weibull function (Equation (A1)), it became rapidly evident that it was inappropriate to describe the depth–number distribution of seeds. $X_0$ was negative in most of the samples, implying the expectation of seeds aboveground somewhere in the air, which is an obvious impossibility. Therefore, the three-parameter Weibull function (Equation (A1)) was replaced by the two-parameter Weibull function (Equation (1)) by setting $X_0 = 0$.

Functions expressed by Equations (1) and (A2) (either using $Y$ or ln $Y$ as dependent variable) could always be fitted to the data of the 20 randomly selected samples.

The adjusted coefficients of determination ($R^2_{aj}$) ranged from 0.987 to ≈1 in the two-parameter Weibull function, from 0.935 to ≈1 in the fourth-degree polynomial using $Y$, and from 0.667 to ≈1 in the fourth-degree polynomial using ln $Y$. Mean $R^2_{aj}$ (±SE) was $0.995 \pm 0.001$ in the Weibull function, $0.986 \pm 0.004$ in the polynomial using $Y$, and $0.901 \pm 0.023$ in the polynomial using ln $Y$. The two-parameter Weibull function ranked first and emerged as preferable to the polynomials, but not very strongly in relation to those using $Y$, clearly more so in relation to the polynomials using ln $Y$.

However, a closer look at the polynomials using untransformed $Y$ showed that only one term was retained in 14 out of 20 equations, meaning that in 70% of the investigated samples the frequency of seeds would be equal in all fractions of soil depth, except in the top one in the two cases where $a_0 \neq 0$ in Equation (A2), which was clearly an impossibility.

Therefore, the fourth-degree polynomial using $Y$ was also rejected and the two-parameter Weibull function was selected to describe the depth–number distributions of spherical and non-spherical seeds in soil.

### Appendix B

In the framework of ongoing research on seed size, the data of the volume and mass of seeds were assembled for 1892 seeds from 15 species, some of them including various cultivars or color morphs. Raw data are available as Supplementary Material (Table S1) in a non-proprietary comma-separated values (CSV) format file.

Volumes were determined from length, width and in 87% of seeds also thickness, using published equations [46]. Measurements of the linear dimensions of seeds were obtained using a digital caliper or, when necessary, a stereomicroscope Leica GZ4 equipped with an eyepiece micrometer Leitz-Periplan 10 × 18 M under 10 × 3 total magnification. Mass was determined by weighing seeds to the nearest 0.1 mg with a scale Precise XR 2055M-DR or with a scale Kern PLJ 600-3 NM.

An allometric equation was fitted by non-linear regression. The resulting equation (Equation (3) above) had a coefficient of determination $R^2 = 0.655$ and was used to estimate seed mass (in mg) from seed volume (in mm$^3$), the latter being equated with mesh size.

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
