# Peer review of "Size–Number and Shape Distribution of Buried Seeds in Soil in a Field Not Cultivated for More Than 10 Years"

_2037-0164, doi:10.3390/ijpb13040039_

Round 1

Reviewer 1 Report (New Reviewer)

This article analyses very well the relationship between size, shape, and depth in the soil of seeds in the study area at a given time. The only point I would make is that this photographs a specific situation that has arisen after more than 10 years of undisturbed soil at a given location. Seeds may have disappeared into the soil because they have been degraded by microorganisms, died, fatal germinated or successfully emerged (as the author said). Are we sure that such a punctual analysis really tells us the “Influence of Size and Shape on Burial of Seeds in Soil?” To understand the influence of size and shape on burying, we would have to eliminate (or consider) all other seedbank decay processes. I appreciate the study, but I would like to read a consideration on this point in the conclusion.

Line 102. Explain better what the author means by “apparently viable seeds”.

Author Response

See pdf file.

Reviewer 2 Report (New Reviewer)

Thank you for the opportunity to provide a review of your manuscript. I have several suggestions that I hope will be useful. 

This paper could be framed within an ecological perspective - some suggestions include:

ecological adaptations - for example - species that are adapted to frequently disturbed soils often produce high numbers of smaller seeds, with the assumption that some portion will be buried too deeply for successful establishment, but that the seeds are frequent enough within the soil profile so that there is always some fraction at depths suitable to germination. Similarly, connecting the species of seeds that were detected with the life history, longevity in soil seed bank, ecology, and plant communities of those species.

It also could be framed and written with a focus on the methodology of quantifying the soil seed bank through seed size, shape, burial depth, and likely transition to seedling stage of the seeds. How do the physical factors involved in seed depth relate to the results?

A few additional suggestions follow. 

Using "lower", "higher", and "smaller" to describe depth is confusing. Consider using deeper and shallower, or greater and lesser instead. Choose one of these pairs and use it consistently throughout.

317-326 - other aspects of the interaction between the seed microenvironment and dormancy include the effect of abrasion, freezing, heating, and light.

Provide an initial description of the size cut offs for small, medium and large classes of seeds, rather than off and on throughout. "Very small" is also used without clearly explaining the actual seed sizes. 

To accompanying the introduction and discussion about how seeds might move upward in the soil seed bank, the equally important explanation about how seeds may move downward should be added. 

Author Response

See pdf file.

Round 2

Reviewer 1 Report (New Reviewer)

The author has addressed my comments and concerns in the revised version. I have no additional comments.

Author Response

See uploaded file.

Reviewer 2 Report (New Reviewer)

Thank you for your letter responding to my comments in the first round of review. I may not have sufficiently communicated the points I was trying to make then. This response intends to make things clearer. The fundamental framing of the purpose, value, or application of this research is still missing. This is the main reason I recommend major revisions. The purpose or rationale is ultimately up to you to as author to determine and communicate to the reader. The suggestions I offered were to frame in in seed ecology, such as more about which strategies (Grimes 1977 R-S-K triangle) would be expected given the land use history of the study site and the species present above ground and in the seed bank. My choice of words may have confounded this. I said "frequently disturbed" when actually "recently disturbed" is more accurate. Under cultivation, the site was frequently disturbed. Frequency and recently are both relative; part of framing the context, rationale, meaning, and relevance of the research is to put it in context and explain if/why the site is considered disturbed or not and what that means for the results you found.

Another option I suggested was to frame it and submit it as a Methods paper, since that part of the manuscript and study design is strong.

Significant and frequent problems with English persist throughout and need to be improved and corrected. I suggest working with a paid service or native speaker to improve the English to a level for publication.  

"Provide an initial description of the size cut offs for small, medium and large classes of seeds, rather than off and on throughout. "Very small" is also used without clearly

explaining the actual seed sizes." I think there was a misunderstanding of what I intended. You have created bins and categorized seeds as very small, smaller, larger, etc. See lines 22-23, 218-220, 269-271 to highlight some of the instances where you refer to the quantitative size and the assigned class size. My comment was to suggest adding text or a brief table that summarizes these cut offs. 

I appreciated the other technical corrections to how depth is described (shallower and deeper, instead of the previous smaller and greater).

Author Response

See uploaded file.

Round 3

Reviewer 2 Report (New Reviewer)

 Related to the rationale, application, and framing of the hypotheses- I think we are each correct, and the disagreement is a failure for me to communicate effectively. My reasons are to offer suggestions to strengthen the relevance of this research, to show the meaning or application that the results have in practice. I think a few sentences, a connecting paragraph that is, would bridge this gap. I hope to explain myself better.

39-44 talks about the relevance of seed banks as their role in the future potential of the above-ground plant community.

45-46 then explains that how deeply or shallowly the seeds are buried plays a large role in determining which seeds are in less or more favorable depths for germination and establishment.

H1 – smaller seeds are buried more deeply than larger seeds

H2 – “differences in shape distributions in soil seed banks are important for seed burial” i.e., we expect a correlation between seed shape and depth of burial (although you don’t specify if the hypothesis is that spherical seeds will be buried more deeply and irregular or oblong seeds will be buried more shallowly, or another specific hypothesis.

This is all good, as is the rest of the introduction except for what is missing: what does it mean, how can plant ecologists (or other relevant specialists) use the results? If the hypothesis that is physical factors (mass and shape) determine how seeds move up and down in the soil profile, how can those results be applied to a different site or system to understand what the potential future above-ground plant community is?

The English has been improved and corrected well throughout since the previous revision. I see what looks to be an error that was overlooked:

6 Abstract: Seeds act as reserves for plant dispersion in time and their burial in soil play^s an essential

“Burial in soil” is a singular subject so the verb should match and be plays rather than play.

Thank you for your responses to the other points I raised in the previous comments. I think you have explained or edited sufficiently.

Author Response

See pdf file.

This manuscript is a resubmission of an earlier submission. The following is a list of the peer review reports and author responses from that submission.

Round 1

Reviewer 1 Report

Dear Authors, kindly check English sentence structure, Grammar errors, throughout the manuscript and suggested modified changes as per the comments given below:

  • Line5-19, the author should write a short methodology for study in the abstract.
  • Line19, the Author should add the statement as a future recommendation.
  • Line 20, These keywords must be removed “burial, depth, and successful germination”
  • Line 34-37, this paragraph needs to be rewritten in a technical way.
  • Line 38, needs to clarify this “Understanding the fate” what does it mean.
  • Line 78, the Author should write coordinate in DDMMSS format for the study area.
  • Line 81, If the land is not cultivated for more than 10 years so how do you maintain the soil fertility for seed grown.
  • Line 139, Figure1,2,3 must improve the higher Dpi as per journal guidelines and also check the other figure throughout the manuscript.
  • The conclusion is not found in the manuscript, Author should write the conclusion for this research manuscript.
  • What are the appendices described, Author suggests adding all appendices as supplementary.

Author Response

Please see PDF file.

Reviewer 2 Report

General comments:

The Author presents a manuscript entitled “Influence of size and shape on burial of seeds in soil", submitted to Soil Systems, MDPI.

The study conducted in the Mitra Experimental Farm near Évora (Southern Portugal), addresses the effect of size and shape on burial seeds of different plant species growing in natural pasture. To address this objective, a “systematic sampling” of soil was conducted on three plots of 6 m x 2 m. Each soil core was divided in fractions of 2.5 cm and sieved through a series of ten different mesh sizes. Seeds were classified for size and shape; no attempt was made to classified seeds species, but those present in all samples. 

Overall, I found the manuscript interesting but challenging. I have read carefully the alternative approach on seed burial, but I have to admit that I am not convinced. Overall, I am not persuaded that smaller spherical seeds are not able to reach depths greater than non-spherical and large seeds (Lines 6-8 “Two hypotheses on depth distribution of seeds in soil were investigated. One, that burial of small seeds is restricted to small depths in comparison to large seeds”). 

I would like to suggest some paper addressing why small spherical seeds are easily reaching greater soil depth than larger seeds (e.g., Saatkamp et al DOI:10.1079/9781780641836.0263; Benvenuti DOI: https://doi.org/10.1017/S0960258507782752; Benvenuti DOI:10.2134/agronj2003.0191).

Furthermore, I am not persuaded that there can be an absence of soil cracks and crevices (Lines 51-52 ”In all likelihood, in the absence of cracks and crevices the same differential retention of seeds occurs in vertical movement in the soil profile. Therefore, we made the hypothesis that in the absence of cracks and crevices the burial of small seeds is restricted to smaller depths than the burial of large seeds.” Cracks and crevices are also driven by the alternation of drought and flood conditions, as well as the alternation of cold and warm weather. How can the Author conceive the absence of cracks and crevices? 

A drawback of the paper is that soil and seeds samples were collected only at one single “time-point”, not representative of the entire dynamic affecting the soil seedbank. The Author presents only a snapshot of the seedbank, not adequate to describe the effect of size and shape on seeds burial, affected in the long-term by biotic and abiotic factors. Furthermore, only three plots of 6m x 2m were observed, and for each plot three samples were collected. 

Is it the size of the sample representative for the study area? How was the area selected? Were the plots similar for vegetation cover and exposure to abiotic factors? Was there any slope difference between plots? Were the plots located at the same distance from agricultural areas that could be representative for insect predation?  What about the seed predation effect? Especially in a no-till system, all seeds laying on the soil surface are affected by predation, which could change the percentage of small seeds on the top soil (easily predated) compared with big seeds. 

Author Response

Please see PDF file.

Reviewer 3 Report

Regarding the way of writing the article, it is important to check that it is written in the third person, in different parts it is referred to in the first person.

The article is a difficult document to read, even more so when there are no explanatory tables or figures.

To perform a more complete work, related information is missing, for example, with the rainfall of the sector, soil humidity, the density characteristics of the plantations, the vegetation present, the type of soil, the management that has been carried out, the requirements of each of the plant species present, among others, because not only the size of the seed is what influences the ease of burial, but also the genetic characteristics of the plant, the texture of the soil (coarser texture, deeper).

Even when the hypothesis is valid in itself, it requires the participation or influence of other factors, which are described in the different articles cited by the researcher.

I believe that it is necessary for the researcher to carry out a more detailed analysis and discussion of the results related to the size and shape of the seed versus sowing depth.

Author Response

Please see PDF file.

Reviewer 4 Report

The author has presented a very common observation in a scientific manner.

Author Response

Please see PDF file.

Round 2

Reviewer 1 Report

I appreciated the improvement of the English level and replies to comments and the information added. But the revised paper has not been improved with the mean changes. In my opinion, the revisions made are not sufficient, therefore I would suggest this paper it is not acceptable in its present form.

Reviewer 2 Report

Dear Luís,

Thank you for the work in revising your manuscript entitled “Influence of size and shape on burial of seeds in soil”. 

I appreciate your effort in critically addressing all the points raised in the previous review, and the information added. 

I am still a bit sceptical about the methodology applied in your work about the systematicity and representativeness of the sampling. 

Just one comment: as you suggest, a blood sample collected from a closed circulatory system is homogeneous, while seed bank is an open heterogeneous system affected by different plant species. 

However, the heterogeneity of the plants in a natural undisturbed system is not absolute but affected by frequency dependence and coexistence. I am assuming that the coexistence of the plant species is patchy, as the system is trying to reach an equilibrium (which it will never be reached!?) 

But I agree with you that this issue is obviously always open to discussion and disagreement. 

Yes, I think it might be useful to put some data available in the appendix for any research interest in plant community dynamics who might want to use these results. 

Reviewer 3 Report

In the introduction it is not understood when it indicates "In small seeds (0.008 mm3; approximately 75% of the total)" and what is the total number of seeds?... 75% of the total seeds evaluated had a size of 0.008 mm3? ...and of this 75% between 52 to 60% were spherical and 76 to 86% were non-spherical? These percentages do not give!!!. What relationship does this have? It must be written clearly.

The number of seeds evaluated is not mentioned, so as to be clear about the percentages of small and large seeds and of these, how many were spherical and non-spherical.

Final conclusions are not written. They must be explicit at the end of the text.